# Fc-Engineered Antibodies with Enhanced Fc-Effector Function for the Treatment of B-Cell Malignancies

**DOI:** 10.3390/cancers12103041

**Published:** 2020-10-19

**Authors:** Hilma J. van der Horst, Inger S. Nijhof, Tuna Mutis, Martine E. D. Chamuleau

**Affiliations:** Department of Hematology, Cancer Center Amsterdam, Amsterdam UMC, VU Medical Center, 1081 HV Amsterdam, The Netherlands; i.nijhof@amsterdamumc.nl (I.S.N.); t.mutis@amsterdamumc.nl (T.M.); m.chamuleau@amsterdamumc.nl (M.E.D.C.)

**Keywords:** antibody therapy, Fc engineering, immunotherapy, B-cell malignancies, B-cell lymphoma, B-CLL, multiple myeloma

## Abstract

**Simple Summary:**

Monoclonal antibody (mAb) therapy has been an important addition to the therapeutic arsenal in B-cell malignancies. MAbs can induce cytotoxicity against B-cell malignancies by antibody effector functions mediated via their fragment crystallizable (Fc) region. In order to enhance the anti-tumor potential of antibodies, various Fc-engineering strategies have been developed. In this review we summarize the well-established as well as recently developed Fc-engineering strategies which are aimed to increase Fc-effector function and to enhance the anti-tumor potency of mAbs. In addition, the increased number of Fc-engineered mAbs in (pre-)clinical development asks for a clear overview describing the specific type of Fc-engineering, their antigen and disease target, and the current developmental stage, which we aimed to provide in this review.

**Abstract:**

Monoclonal antibody (mAb) therapy has rapidly changed the field of cancer therapy. In 1997, the CD20-targeting mAb rituximab was the first mAb to be approved by the U.S. Food and Drug Administration (FDA) for treatment of cancer. Within two decades, dozens of mAbs entered the clinic for treatment of several hematological cancers and solid tumors, and numerous more are under clinical investigation. The success of mAbs as cancer therapeutics lies in their ability to induce various cytotoxic machineries against specific targets. These cytotoxic machineries include antibody-dependent cellular cytotoxicity (ADCC), antibody-dependent cellular phagocytosis (ADCP), and complement-dependent cytotoxicity (CDC), which are all mediated via the fragment crystallizable (Fc) domain of mAbs. In this review article, we will outline the novel approaches of engineering these Fc domains of mAbs to enhance their Fc-effector function and thereby their anti-tumor potency, with specific focus to summarize their (pre-) clinical status for the treatment of B-cell malignancies, including chronic lymphocytic leukemia (CLL), B-cell non-Hodgkin lymphoma (B-NHL), and multiple myeloma (MM).

## 1. Introduction

Naturally, antibodies (Abs) are produced by B-cells as a polyclonal population, with high specificity for their distinct target antigen and epitope. Antibodies thereby play various important roles in our immune system. The field of therapeutic Abs commenced in 1975, when the development of the mouse hybridoma technology enabled the production of large amounts of murine monoclonal (m) Abs [1]. However, murine mAbs elicited an immunogenic response in human patients. To reduce this immunogenicity, chimeric mAbs, consisting of a constant human domain fused to a variable mouse domain, were developed [2]. The chimeric mAb rituximab targeting cluster of differentiation (CD) 20 was the first FDA-approved mAb for cancer therapy in 1997. The development of advanced design technologies such as human antibody gene expression libraries and transgenic animals allowed the engineering of humanized (the hypervariable region of a murine antibody grafted in a human antibody) and fully human mAbs [3]. To be successfully applied in the clinic, mAbs generally require additional engineering to improve their affinity, limit any biophysical liabilities, and to increase their half-life. Currently, 30 mAbs are clinically approved for treatment of cancer, and this number is rapidly increasing: in the last decade, the number of mAbs that have entered late-stage clinical studies has been tripled [4]. The therapeutic potential of mAbs has been exploited by the development of antibody fusion products, such as bispecific antibodies or antibody drug conjugates, which take advantage of specific antigen binding properties of antibodies to precisely target cytotoxic cells or toxic agents to cancerous cells. A novel development in antibody engineering is the modification of the antibody fragment crystallizable (Fc) region in order to increase the Fc tail-mediated effector functions, including antibody-dependent cellular cytotoxicity (ADCC), antibody-dependent cellular phagocytosis (ADCP), and complement-dependent cytotoxicity (CDC), to induce tumor cytotoxicity more effectively. Numerous Fc-engineered antibodies have demonstrated clinical activity or are under preclinical investigation.

In this review, we will outline the novel approaches of engineering Fc domains of mAbs to enhance their Fc-effector function and anti-tumor potency, with specific focus to their (pre-) clinical status for the treatment of B-cell malignancies, including chronic lymphocytic leukemia (CLL), B-cell non-Hodgkin lymphoma (B-NHL), and multiple myeloma (MM). Not described in this review is the application of Fc engineering in order to improve antibody half-life, to silence mAb effector functions in case of antibodies used as receptor agonists or antagonists or as drug delivery vehicles, and to increase the direct, not Fc-effector function-mediated, anti-tumor potency of mAbs.

## 2. Antibody Structure

Antibodies are mono- or polymers of immunoglobulins (Ig) consisting of two identical pairs of heavy (H) and light (L) chains, which are linked through non-covalent interactions and disulfide bonds to form a Y-shaped structure [5]. All H and L light chains contain a single variable domain (V_L_), which also consists of hypervariable regions. The combination of the (hyper) variable regions of the H and L chains determines the antigen specificity and affinity of an antibody. The L chains contain a single constant (C_L_) domain to make a stable link with the H chain. The number of constant domains of the H chain (C_H_) is dependent on the isotype of the antibody: IgA, IgD, and IgG contain three (C_H_1–3), and IgE and IgM contain four constant domains (C_H_1–4). The first C_H_ is linked with C_L_ to the variable regions, which together form the fragment antigen binding (Fab) region. The heavily glycosylated C_H_2–3 or C_H_2–4 domains are linked to C_H_1 via a flexible hinge region and constitute the Fc region (Figure 1A). The engineering of this region, which is responsible for the isotype- and subclass-dependent Fc-mediated effector functions of antibodies, will be the main focus of this review.

## 3. Fc-Effector Functions

Upon antigen engagement, IgG antibodies can induce direct anti-tumor effects via triggering the cell death signaling pathways and via blockade of essential receptor systems, as well as indirect anti-tumor effects via their Fc-mediated effector functions, by engaging other immune cells or killer mechanisms. The Fc-mediated effector functions of antibodies include antibody-dependent cellular cytotoxicity (ADCC), antibody-dependent cellular phagocytosis (ADCP), and complement-dependent cytotoxicity (CDC), and have been shown to be crucial for the therapeutic efficacy of most clinically approved antibodies (Figure 1B). Among the four IgG subclasses, IgG1 and IgG3 induce the strongest Fc-effector functions [6]. However, since IgG1 has the longest half-life and is more stable than IgG3 [7], most therapeutic antibodies with Fc-mediated functions are of IgG1 isotype.

### 3.1. ADCC/ADCP

The IgG1-induced ADCC and ADCP response is mediated via binding to Fc gamma receptors (FcγR), which are expressed on innate effector cells, including monocytes, monocyte-derived cells, basophils, mast cells, and natural killer (NK) cells. The FcγR family consists of the activating FcγRI (CD64), FcγRIIa (CD32a), FcγRIIIa (CD16a), and FcγRIIIb (CD16b), and the inhibitory FcγRIIb (CD32b). Of all FcγRs, only FcγRI, which plays a major role in myeloid cell activation, is classified as a high-affinity Fc receptor [8]. All other FcγRs require binding of multivalent IgG-antigen immune complexes in order to provide sufficient avidity to activate downstream signaling and induce antibody-mediated ADCC or ADCP [9].

Natural killer (NK) cells are considered the most potent inducers of ADCC. NK cells as well as monocytes and macrophages express FcγRIIIa, however, only NK cells exclusively express FcγRIIIa [10]. Triggering of FcγRIIIa induces phosphorylation of immunoreceptor tyrosine-based activation motifs (ITAMs), which activates a downstream signaling cascade resulting in the release of cytotoxic granules containing perforin and granzyme in the immune synapse formed between the NK cell and the target cell, leading to target cell death [11].

In contrast to NK cells, the phagocytic monocytes, neutrophils, and macrophages co-express activating and inhibitory FcγRs. ADCP can be induced by various activating FcγRs including FcγRIIIa (CD16a), FcγRIIa (CD32A), and FcγRIIIb (CD16b), which all signal via ITAMs, and may be inhibited by FcγRIIb (CD32b), which signals via immunoreceptor tyrosine-based inhibitory motifs (ITIMs). The balance of activating and inhibitory signaling dictates ADCP induction, which occurs via internalization and degradation of the antibody-opsonized target by the phagocyte [12].

FcγRs are highly polymorphic, their genes are known to have several single-nucleotide polymorphisms (SNPs). Such polymorphisms can affect the affinity of the FcγR for Ig molecules [13]. Both FcγRIIa and FcγRIIIa are known to exist as two allotypic variants, which are associated with clinical response for therapeutic mAbs. In the extracellular domain of FcγRIIa, a C > T substitution at amino acid position 131 results in a histidine (131R) to arginine (131R) replacement [14]. FcγRIIa binds with high affinity to IgG1 and IgG3. The amino acid position 131 is polymorphic for IgG2 binding: the 131H variant can bind to IgG2 with high affinity, while 131R barely binds to IgG2 [13]. For FcγRIIa, a T > G substitution can occur at amino acid position 158 resulting in valine (158V) to phenylalanine (158F) replacement. The FcγRIIIa-158V variant was shown to bind to IgG1 and IgG3 with a higher affinity compared to FcγRIIIa-158F [15].

The FcyRIIa (131H/R) and FcyRIIIa (158F/V) polymorphisms are associated with clinical response for several clinical-approved mAbs including rituximab, trastuzumab, and cetuximab [16,17,18].

### 3.2. CDC

Therapeutic IgG1 antibodies can activate the classical pathway of the complement system by binding of the Fc region to the complement protein C1q, which initiates a cascade of proteolytic cleavage steps. This results in formation of the membrane attack complex (MAC), consisting of the complement products C5b to C9. The MAC generates pores in the cell membrane which initiates target cell lysis, termed complement-dependent cytotoxicity (CDC) [19]. Similar to FcγR, the affinity of C1q for IgG-Fc is low and binding is dependent on multivalent IgG-antigen immune complexes to provide sufficient avidity [20,21,22]. The ability of IgG1 antibodies to activate the complement pathway is highly dependent on antigen density, size, and fluidity [23,24,25]. High-resolution crystallography studies have recently revealed that, dependent on such factors, specific non-covalent interactions between IgG Fc domains induce ordered hexamer formation (mAb hexamerization) on the cell surface that provides a docking platform for the six-globular-headed C1q molecule and thereby efficiently activates the complement pathway [26].

## 4. Fc Engineering to Enhance Fc-Effector Functions

The hinge and the proximal C_H_2 regions of the Fc tail are considered critical for Fc interaction with FcγRs and C1q. The interface of these regions contains the binding sites, while the structural conformation of the C_H_2 domain allows engagement of C1q or FcγR. In addition, the C_H_2 domains are post-translationally modified by asparagine(*N*)297-linked glycosylation, and glycosylation and the specific glycan composition contribute to the stability and the dynamics of the C_H_2 domains [27,28,29]. Glycan components include core units of N-acetylglucosamine (GlcNAc) and mannose, with additional variations in galactose, bisecting GlcNAc, fucose, and sialic acid.

Detailed understanding of Fc interactions with C1q and FcγR opened up opportunities to modulate C1q and FcγR binding by Fc engineering. In order to enhance ADCC, ADCP, and CDC, studies have employed site-directed mutagenesis (sequence variations), Fc glycosylation modification (glycoengineering), and avidity modulation, which will be outlined below.

### 4.1. Enhancing ADCC

#### 4.1.1. Glycoengineering to Enhance FcγR Affinity

Fc glycosylation is required for binding the low-affinity FcγR [30,31]. Generally, aglycosylation is thought to completely abrogate FcγR effector functions [32,33], but several aglycosylated Fc variants with intact FcγR effector function have been reported [34,35,36]. Altering the specific composition of the Fc glycan can increase the affinity for FcγR. The removal of core fucose (afucosylation) has been shown to highly increase FcγRIIIa binding affinity and consequently increase ADCC [37,38] (Figure 2A). This effect has been attributed to an interaction between the Fc-glycan and the N-glycan attached to Asn 162 of the FcγRIIIa [39], however, the exact nature of the interaction is still debated. It has been suggested that core fucose restricts the number of conformations recognized by the FcγRIIIa N-glycan [40], while others suggest that core fucose inhibits direct carbohydrate–carbohydrate interactions with the receptor glycan [41,42]. Nevertheless, afucosylation is widely accepted as an effective approach to increase the potency of IgG1 antibodies to induce ADCC.

To a lesser extent, Fc galactosylation is also suggested to modulate FcγRIIIa binding. The reported effects of hypergalactosylation on ADCC, however, range from completely absent to positive without addition of afucosylation [38,43,44] and positive with addition of afucosylation [45,46]. These large variations in study results might be explained by differential interactions of the galactose on the different N-glycan arms with FcγR [47]. The effects of Fc sialylation on ADCC have been described to be minimal, and completely absent in addition to afucosylation [43,46,48].

#### 4.1.2. Site-Directed Mutagenesis to Enhance FcγR Affinity

High-resolution structural Fc analysis revealed the specific FcγR-binding sites, which has laid the foundation for structure-guided identification of affinity-enhancing mutations. FcγRs interact with residues Leu234–Ser239 on the lower hinge and residues Asp265–Glu269, and Asn297–Thr299 on the C_H_2 domain [49,50]. Since then, it has become clear that numerous positions within or in close proximity to this region can be mutated to improve FcγR binding affinity. Alanine screening in the C_H_2 and C_H_3 domains revealed that several mutations could enhance binding to FcγRIIIa, with the most potent mutations combined in S298A/E333A/K334A for enhanced ADCC [51]. A study using computational design algorithms and high-throughput screening demonstrated that S239D/I332E mutations could also enhance FcγRIIIa binding and ADCC [52]. Both S298A/E333A/K334A and S239D/I332E highly enhanced binding for the lower-affinity polymorphic variant (F158) of FcγRIIIa [51,52]. The P247I/A339Q mutations, which were applied in the anti-CD20 mAb ocaratuzumab, have also been shown to enhance binding to the lower-affinity FcγRIIIa [53] (Figure 2B).

Structural analyses of the Fc-FcγR interaction have revealed that the Fc binding to FcγR is asymmetrical: the receptor binds to different residues on each Fc domain. Hence, it seemed likely that applying mutations to Fc regions asymmetrically could maximize the FcγR binding affinity. Indeed, Fc heterodimeric antibodies improved C_H_2 domain stability and the consequent FcγRIIIa binding as compared to a symmetrically mutated Fc variant [54,55]. In addition, afucosylation of the heterodimeric antibodies further improved the FcγRIIIa binding [55].

#### 4.1.3. Fc Multimerization

In comparison to affinity modulation, avidity modulation is a less established but more straightforward approach. Fc duplication (or tandem-Fc) or multiplication, whereby multiple Fcs are linked within one IgG1 molecule, has been shown to augment FcγR binding avidity and increase ADCC and ADCP [56,57,58,59] (Figure 2C). A theoretical safety concern for Fc multiplication strategies is the fact that natural antibody oligomerization may result in unwanted immune activation [60]. However, studies have reported minimal in vitro aggregation and no in vivo adverse events so far [57,59].

### 4.2. Enhancing ADCP

#### 4.2.1. Glycoengineering to Enhance FcγR Affinity

Strategies enhancing ADCC via increased affinity for FcγRIIIa on NK cells can also enhance ADCP via increased antibody binding to monocytes and macrophages, since these cells also express FcγRIIIa. ADCP induced by neutrophils can be improved as well since neutrophils express FcγRIIIb, which shares 97% sequence homology with FcγRIIIa. It has indeed been demonstrated that afucosylated mAbs can induce higher levels of ADCP [61,62].

#### 4.2.2. Site-Directed Mutagenesis to Enhance FcγR Affinity

Different than ADCC, ADCP induction is highly dependent on the balance of binding to the activating receptors versus the inhibitory receptor FcγRIIb. The activating FcγRIIa shares 90% similarities with the inhibitory FcγRIIb [63]. Hence, selectively increasing FcγRIIa binding without influencing or while even decreasing the inhibitory FcγRIIb binding remains a great challenge in enhancing ADCP and requires more careful engineering. Increasing FcγRIIa binding while simultaneously decreasing FcγRIIb was achieved by mutations F243L/R292P/Y300L/V305I/P396L [64]. In another study, in which ADCC could be enhanced by S239D/I332E mutations, a third mutation (A330L) was necessary to improve ADCP because the sole S239D/I332E mutation also resulted in increased binding to FcγRIIb [52]. Another study identified the G236A mutation to selectively enhance FcγRIIa binding. This study demonstrated that the addition of G236A to S239D/I332E and S239D/A330L/I332E resulted in enhanced ADCP, in addition to the improvement of ADCC [65,66] (Figure 2B).

#### 4.2.3. Fc Multimerization

Fc multimerization strategies are not FcγR-specific. Therefore, such strategies will enhance the binding of mAbs to other low-affinity FcγRs, including to the inhibitory FcγRIIb. Nonetheless, it appeared possible to increase FcγRIIa binding and ADCP by Fc multimers [57,67]. However, since binding to the inhibitory FcγRIIb was also increased [57], Fc multimerization strategies might require further Fc engineering to optimally enhance the ADCP.

### 4.3. Enhancing CDC

#### 4.3.1. Glycoengineering to Enhance C1q Binding Affinity

While afucosylation significantly enhances ADCC and ADCP by facilitating the interaction with the FcγRIIIa glycan, it minimally affects CDC [46]. Sialyation seems to have moderate effects on C1q binding. Some studies reported increased and some others reported decreased C1q binding by sialyation [46,68,69]. Instead, galactose is the key glycan for C1q binding. Numerous studies demonstrated enhanced C1q binding and CDC by Fc galactosylation [46,69,70,71] (Figure 2A). Molecular interactions between galactose and amino acid residues on the C_H_2 domains possibly increase C1q binding affinity [72]. It has also been suggested that Fc glycosylation modulates Fc/Fc interactions and thereby affects not the affinity but the avidity of C1q binding [73].

#### 4.3.2. Site-Directed Mutagenesis to Enhance C1q Binding Affinity

The first structural analysis studies revealed that the residues D270, K322, P329, and P331 of the C_H_2 domain were critical for the interaction with C1q [74,75]. More recently, it has been shown that there are two main interaction sites: residues 266–272 and 294–300 on one C_H_2 domain and residues 325–331 on the other [76]. Mutations in residues located on or in proximity to these binding sites significantly affected C1q binding: the double mutant K326W/E333S and triple mutant S267E/H268E/S324T enhanced C1q binding and CDC [77,78] (Figure 2B). The hinge region also plays a role in complement activation, because this region affects the flexibility of the Fc tail, thereby determining the ability to fix C1q. Indeed, certain mutations in the upper hinge region could enhance C1q binding and CDC [79].

#### 4.3.3. Antibody Hexamerization to Facilitate C1q Binding

In addition to affinity modulation, site mutagenesis can also be performed in order to modulate avidity. Proceeding from the finding that antibody hexamers facilitate C1q binding, the essential first step in CDC, a novel strategy was developed in order to improve the hexamer forming of antibodies upon target antigen binding. Introducing the specific point mutations E345R and E430G at the Fc and C_H_2-C_H_3 interface could indeed stimulate the Fc/Fc interactions between antibodies and facilitate the natural concept of antibody hexamerization, leading to superior C1q binding and enhanced CDC [26,80] (Figure 2B,E). Since Fc hexamerization by these specific point mutations only occurs upon antigen binding on the cell surface, antibodies generated by this so called “HexaBody” technology retain the pharmacokinetics of conventional IgG1 antibodies.

#### 4.3.4. Cross-Isotype Antibodies

Although both IgG1 and IgG3 can effectively activate complement, IgG3 antibodies can bind C1q more effectively [81]. Therefore, cross-isotype antibodies have been generated by replacing the C_H_2 and C_H_3 domains of an IgG1 antibody with the corresponding regions of an IgG3 antibody, which increases the CDC response [82] (Figure 2D).

## 5. Generation of Fc-Engineered mAbs

Although the hybridoma technology revolutionized the field of therapeutic antibodies, most mAbs that are currently approved for therapeutic use are generated by mammalian expression systems, which allow higher antibody yields and preserve post-translational modifications, generating a higher-quality mAb product. Mammalian expression systems often use the variable regions derived from the hybridoma or phage display technologies. The sequence of the desired region is cloned into the appropriate expression vector and subsequently transfected into the expression system. Currently available mammalian expression systems include various Chinese hamster ovary (CHO) cell lines, mouse myeloma (NS0), and mouse hybridoma (Sp2/0) cell lines. In addition, several human expression systems are available, including embryonic kidney (HEK293), amniotic (CAP), a hybrid of HEK293 and lymphoma (HKB-11), and embryonic retina (PER.C6). The human expression systems, however, provide transient expression and are therefore only suitable for preclinical purposes.

The current approaches of antibody sequence engineering at the Fc site apply site-directed mutagenesis either directly once heavy and light chain sequences are available (for structure-based sequence engineering) or to generate large phage or yeast display libraries to screen for the most optimal Fc variant (empirical-based sequence engineering). Glycoengineered antibodies require more complex adaptations in the manufacturing protocol, which will be outlined below.

### Glycoengineered mAbs

Mammalian expression systems allow conventional post-translational modifications and can also be modified to alter specific post-translational modifications, such as Fc glycosylation. However, glycosylation is a complex process and cannot be controlled completely as cell culture conditions can alter the glycosylation pattern [83,84]. To create glycoengineered antibodies in order to develop antibodies with improved ADCC activity, several modified mammalian expression systems were developed. Double knockout of the enzyme α1,6-fucosyltransferase 8 (FUT8), which catalyzes the transfer of fucose from GDP-fucose to N-acetylglucosamine (GlcNAc), in CHO cell lines resulted in the production of afucosylated antibodies [85]. Alternatively, CHO cells were engineered to express β(1,4)-*N*-acetylglucosaminyltransferase III (GnTIII). IgGs produced by mammalian cells have very low or no bisecting GlcNAc, in contrast to IgGs present in human serum, and increasing the number of bisecting GlcNAc improved ADCC levels [86,87]).

## 6. Clinical Experience with Fc-Engineered mAbs for B-Cell Malignancies

Although various strategies can enhance ADCC, ADCP, or CDC effector function, they do not uniformly increase these effector functions when applied to different antigens, since antigen binding also affects the C1q and FcγR binding via structural allostery [88,89]. In addition, there is a partial overlap in the Fc-binding sites for C1q and FcγR [6]. Therefore, modulating the Fc tail to enhance ADCC/ADCP can negatively influence CDC, and vice versa. It is thus recommended to evaluate Fc-effector function-enhancing strategies for each target individually.

In B-cell malignancies, a wide variety of disease-associated targets are available, including various lineage-specific surface molecules. The Fc-engineered mAbs for these target antigens are discussed below for each relevant disease subtype (Table 1).

### 6.1. B-CLL and B-NHL

#### 6.1.1. CD20

CD20 is expressed on almost all healthy and malignant B-cells, but is not expressed by precursor B-cells and plasma cells, making it the ideal therapeutic target for B-cell malignancies [90]. The CD20-targeting chimeric mAb rituximab was the first mAb to be approved by the FDA for cancer therapy in 1997 and is currently still part of the first line of immune-chemotherapy regimens for patients with B-NHL and CLL. Although rituximab is capable of both ADCC/ADCP and CDC induction, multiple Fc engineering strategies have been explored to enhance the effector functions of CD20-targeting mAbs, either type I or type II. The CD20 mAbs are classified as type I and II based on their ability to reorganize the CD20 molecules into lipid rafts. Type I CD20 mAbs, such as rituximab, can induce CD20 reorganization and efficiently activate the complement pathway, whereas type II CD20 mAbs are poor complement activators but instead induce direct cell death. Both type I and II mAbs can induce ADCC [91,92].

Glycoengineered CD20-targeting mAbs include obinutuzumab (GA101) and ublituximab (TG-1101). Obinutuzumab is a type II glycoengineered (non-fucosylated) humanized anti-CD20 IgG2 mAb which targets a different but overlapping epitope on CD20 compared to rituximab [93]. In comparison to rituximab, a significant clinical benefit of obinutuzumab was observed for FL and CLL in combination with chemotherapy [94,95,96], and obinutuzumab has received FDA approval for FL and CLL. Obinutuzumab in combination with CHOP (G-CHOP) did not show a PFS benefit compared to R-CHOP for treatment-naïve DLBCL [97,98]. Ublituximab is a type I glycoengineered (low-fucose content) chimeric anti-CD20 IgG1 which targets a unique epitope on CD20 and is currently under clinical investigation. Ublituximab increased the ADCC of CLL cells in vitro and ex vivo compared to rituximab [99,100], and induced ADCC in rituximab-resistant B-NHL in in vitro and in vivo models [101]. Ublituximab has shown promising phase 2 and 3 clinical efficacy either as a single agent or in combination with ibrutinib and umbralisib, the first BTK inhibitor and a next-generation PI3K inhibitor, in high-risk CLL and B-NHL patients [102,103,104]. In addition, several clinical trials are ongoing, including trials investigating the efficacy in treatment-naïve FL and in progressive CLL (ClinicalTrials.gov Identifier: NCT03828448 and NCT04149821, respectively) and a trial investigating the combination of ublituximab with an anti-PDL1 mAb (TG-1501) (ClinicalTrials.gov Identifier: NCT02535286).

The relevance of FcγRIIIa polymorphisms for antibodies targeting CD20 has been demonstrated by the higher response rates of rituximab in patients with the 158V variant [16,105,106]. Fc-mutated CD20-targeting mAbs that were clinically evaluated and designed to enhance affinity for the low-affinity variant FcγRIIIa-158F include the humanized mAbs ocaratuzumab (AME-133v; LY2469298), PRO131921 (RhuMAb v114), and ocrelizumab. Ocaratuzumab was generated by screening for Fc modifications that enhance ADCC, which led to the identification of the P247I/A339Q mutations that enhanced binding to both allelic variants of FcγRIIIa, in addition to Fab modifications that enhance antigen binding [107]. In vitro, ocaratuzumab induced ADCC in CLL cells at higher levels than rituximab, and similar levels to obinutuzumab [53]. A phase 1/2 clinical trial demonstrated the activity and tolerability of ocaratuzumab in previously treated FL patients with low-affinity FcγRIIIa [108,109]. PRO131921 is Fc-modified (unspecified) to enhance C1q binding in addition to enhanced FcγRIIIa binding, and was demonstrated to enhance ADCC and CDC in vitro compared to rituximab. A phase 1 trial of PRO131921 in relapsed and/or refractory follicular lymphoma patients who previously received rituximab showed tolerability [110]. However, the clinical development of both ocaratuzumab and PRO131921 has been discontinued [111], no information regarding the reason has been disclosed. Ocrelizumab demonstrated activity in a phase 1–2 trial in patients with relapsed/refractory follicular lymphoma [112], but is, at the moment, only registered for the treatment of patients with multiple sclerosis.

Several other Fc-engineered CD20-targeting antibodies have been explored in preclinical studies. The strong CDC induction of IgG3 antibodies targeting CD20 [113,114] favors the development of IgG1/IgG3 isotype variants, and a CD20-targeting afucosylated IgG1/IgG3 isotype variant, with increased CDC and ADCC levels in vitro [115]. Applying multiple Fc-enhancing strategies simultaneously also proved beneficial for a nonfucosylated rituximab variant containing the S267E/H268F/S324T/G236A/I332E mutations, which enhanced both ADCC and CDC in vitro [116].

#### 6.1.2. CD37

Similar to CD20, CD37 is expressed on all mature B-cells, but absent or expressed at very low levels on stem cells, precursor B-cells, and plasma cells [117,118]. Several CD37-targeting therapeutics have been clinically evaluated, including several immunoconjugates but also two Fc-engineered antibodies. BI 836,826 (MAb 37.1) is an Fc-mutated (S239D/I332E) chimeric IgG1 with enhanced ADCC in addition to pro-apoptotic activity. BI 836,826 demonstrated potent cytotoxicity in CLL cells ex vivo, especially in combination with the PI3K inhibitor idelalisib in relapsed CLL [119,120]. In phase 1 clinical trials in relapsed/refractory CLL and relapsed/refractory B-NHL, acceptable tolerability and preliminary efficacy was observed [121,122]. However, a phase 1b/2 trial of BI 836,826 in combination with gemcitabine and oxaliplatin in DLBCL was halted prematurely due to dose-limiting toxicities (DLTs) (ClinicalTrials.gov Identifier: NCT02624492). BI 836,826 has been discontinued from further clinical development. The Fc-engineered DuoHexaBody-CD37 is a biparatopic (dual-epitope-targeting) CD37-targeting IgG1 antibody with the E430G hexamerization-enhancing mutation that induces potent CDC, in contrast to native CD37-targeting antibodies [123]. DuoHexaBody-CD37 showed ex vivo efficacy in B-CLL and various B-NHL (van der Horst et al. [124]), and a first-in-human clinical trial has recently been initiated (ClinicalTrials.gov Identifier: NCT04358458).

#### 6.1.3. BAFF-R

B cell–activating factor (BAFF) is an immunomodulatory cytokine which regulates B-cell survival and activation. BAFF can bind to three receptors although only one of them binds BAFF with high specificity: the BAFF receptor (BAFF-R) [125]. BAFF-R is expressed on almost all normal and malignant B-cells, but not on pre-B-cells, and is therefore considered an appropriate target for B-CLL and B-NHL. Ianalumab (VAY736; B-1239) is a fully human BAFF-R-targeting glycoengineered (afucosylated) IgG1 antibody. Although ianalumab also blocks receptor signaling and proliferation, ADCC induction mediated by the afucosylated Fc domain was demonstrated to be crucial for potent cytotoxicity. Furthermore, ianalumab induced higher levels of ADCC in CLL cells than rituximab and the Fc-engineered obinutuzumab, and combining ianalumab with ibrutinib could further enhance efficacy in vivo [126,127]. A phase 1 clinical trial is currently active to evaluate ianalumab in combination with ibrutinib for CLL patients (ClinicalTrials.gov Identifier: NCT03400176).

#### 6.1.4. CD19

CD19 expression is restricted to the B-cell lineage but is in contrast to CD20 also expressed on precursor B-cells. CD19 is highly expressed in B-NHL and several leukemias, including CLL and ALL. In addition, although CD19 is generally considered to be absent on plasma cells, it has been shown that some multiple myeloma (MM) cells express CD19 at extremely low density which might suffice for targeted therapy [128]. Unmodified CD19-targeting antibodies induce limited ADCC/ADCP and CDC, partly because they are rapidly internalized [129]. CD19 is therefore mostly used as a target for T-cell engagers, such as bispecific antibodies or chimeric antigen receptors (CARs), but some Fc-engineered CD19-targeting antibodies are also clinically evaluated. The CD19-targeting afucosylated mAbs inebilizumab (MEDI-551) and MDX-1342 and the Fc-mutated (S239D/I332E) mAb tafasitamab (MOR208; XmAb5575) all enhance ADCC levels in vitro relative to native CD19 mAbs [130,131,132]. Inebilizumab was tested in phase 1 trials and showed tolerability and preliminary efficacy in CLL, FL, DLBCL, and MM [133]. However, phase 2 trials of inebilizumab in combination with chemotherapy in CLL and DLBCL did not show any significant differences in outcome compared to rituximab in combination with chemotherapy (ClinicalTrials.gov Identifier: NCT01466153; ClinicalTrials.gov Identifier: NCT01453205). A phase 1 study of inebilizumab in relapsed or refractory advanced B-cell malignancies has recently been completed (ClinicalTrials.gov Identifier: NCT00983619). MDX-1342 has been tested in a phase 1 study in in CLL patients (ClinicalTrials.gov Identifier: NCT00593944), but the study was halted prematurely and the program has been discontinued without further disclosure. Tafasitamab was demonstrated safe and efficacious in a phase 1 trial in relapsed CLL and a phase 2 trial in relapsed and refractory B-NHL [134,135]. Moreover, in vitro studies suggested that lenalidomide can further enhance the ADCC effects of tafasitamab, and the combination with lenalidomide resulted in a high response rate of relapsed and refractory DLBCL patients. [136]. Tafasitamab has been granted accelerated FDA approval in combination with lenalidomide for patients with relapsed DLBCL.

### 6.2. Multiple Myeloma (MM)

#### 6.2.1. CD38

CD38 is an attractive target for multiple myeloma due to its high and uniform expression on MM cells, while its expression on myeloid and lymphoid cells and in non-hematopoietic tissue is relatively low. The unmodified CD38-targeting antibody daratumumab received FDA approval in 2019 and induces MM cell cytotoxicity via ADCC, ADCP, and CDC in addition to direct cell death [137]. To further increase the CDC potential of CD38-targeting antibodies, the Fc-engineered antibody HexaBody-CD38 (GEN3014) carrying the E430G hexamerization-enhancing mutation has been developed. HexaBody-CD38 demonstrated superior CDC activity in vitro compared to daratumumab and showed promising anti-tumor activity in vivo [138]. In addition, the Fc multimerization technology has been employed to generate the anti-CD38 selective immunomodulator of the Fc receptor antibody (SIFbody), with enhanced binding to the Fcγ receptors and C1q resulting in CDC activity and NK- and macrophage-mediated killing in vitro. The anti-CD38 SIFbody also demonstrated increased efficacy ex vivo compared to daratumumab [139].

#### 6.2.2. HM1.24

HM1.24 was first described to be preferentially overexpressed on normal and malignant plasma cells [140,141], although more recent studies also demonstrated HM1.24 expression on B-CLL and lymphoma and several solid tumors [142,143,144,145,146]. Antibodies targeting HM1.24 for MM exhibited in vitro and in vivo anti-tumor activity. However, a phase 1 study of the humanized anti-HM1.24 unmodified antibody AHM in relapsed/refractory MM could not demonstrate significant efficacy. Glycoengineered (afucosylated) variants of AHM and the Fc-mutated (S239D/I332E) anti-HM1.24 antibody XmAb5592 could enhance ADCC as well as ADCP compared to AHM in preclinical studies, and warrant further clinical testing [147,148,149].

#### 6.2.3. ICAM-1

The intercellular adhesion molecule-1 (ICAM-1/CD54) mediates adhesion of MM cells to bone marrow stromal cells (BMSCs). CD54 is highly expressed on MM cells and associated with advanced disease stage and resistance to chemotherapy, which makes ICAM-1 an interesting target for MM [150,151]. The unmodified ICAM-1-targeting antibody BI-505 induced potent anti-myeloma activity in vitro and in vivo, which was predominantly macrophage-mediated [152]. BI-505 progressed to clinical trials, and although a phase I trial demonstrated good tolerability, BI-505 lacked significant efficacy in a phase II trial in MM [153,154]. To potentially enhance efficacy in vivo, Fc engineering (S239D/I332E) has been applied to the anti-ICAM-1 fully human IgG1 antibody MSH-TP15, which binds to a distinct but overlapping epitope compared to BI-505, with enhanced ADCC and ADCP activity in vitro and improved tumor control in vivo compared to its unmodified counterpart [155,156].

#### 6.2.4. BCMA

B-cell maturation antigen (BCMA; CD269; TNFRSF17) plays a significant role in the differentiation of B-cells to plasma cells and is required for plasma cell longevity [157]. BCMA expression is specific for plasma cells and MM cells even overexpress BCMA [158]. Multiple T-cell engagers targeting BCMA are currently being clinically evaluated and expected to receive approval for clinical application soon. SEA-BCMA is a glycoengineered (afucosylated) humanized BCMA-targeting IgG1 antibody and showed promising preclinical activity via induction of ADCC and ADCP as well as a block in proliferation [159]. SEA-BCMA is currently evaluated in a phase 1 safety study in relapsed/refractory MM patients (ClinicalTrials.gov Identifier: NCT03582033).

## 7. Conclusions and Future Perspective

In this review, we have illustrated various strategies to enhance Fc-mediated effector functions and we have summarized their clinical application in chronic lymphocytic leukemia (CLL), B-cell non-Hodgkin lymphoma (B-NHL), and multiple myeloma (MM). Fc-mediated effector functions are being enhanced to increase their anti-tumor potency or when a specific effector function is beneficial, i.e., improving ADCC induction for combination therapy with lenalidomide or in patients with low-affinity FcγR polymorphisms. In addition, Fc engineering can be applied to antibodies which depend on antibody clustering or FcγR-mediated antibody crosslinking for agonism of receptors, such as antibodies targeting the costimulatory protein CD40 or antibodies targeting death receptors 4 or 5, and thus benefit from similar strategies as discussed here [160,161,162].

Generally, mAbs that are Fc-engineered to improve their effector functions are capable of enhancing in vitro and in vivo anti-tumor potency compared to their parental unmodified mAb. Various Fc-engineered mAbs also demonstrated clinical efficacy, and are already approved for clinical use. However, other Fc-engineered mAbs demonstrated toxicity in clinical trials or failed to induce significant clinical efficacy, and were discontinued for development. Increasing the clinical success of Fc-engineered mAbs requires more empirical in vitro/in vivo screening to determine the most favorable Fc engineering strategy or combination of strategies. In addition, understanding (i) the specific contribution of ADCC, ADCP, and CDC to the clinical efficacy of mAbs in hematological malignancies and (ii) the exact clinical effect of the different Fc engineering strategies could allow for developing Fc engineering strategies customized to a specific target and disease. A step forward in understanding the specific contribution of mAb effector functions is the recent development of Fc engineering strategies that enhance CDC specifically, such as the HexaBody technology. Until recently, clinically evaluated mAbs were mostly Fc-engineered to enhance ADCC function. Hence, evaluating Fc-engineered mAbs with enhanced CDC and potentially comparing them to mAbs with enhanced ADCC could provide crucial information regarding the contribution of mAb effector functions to clinical efficacy and toxicity, and the results of such clinical trials are highly anticipated.

To conclude, our advanced knowledge of Fc structure and Fc-mediated effector function has enabled the clinical development of Fc-engineered mAbs. Expanding our clinical experience with these Fc-engineered mAbs will provide valuable information that could allow the development of antibodies with tailor-made effector functions.

## Figures and Tables

**Figure 1 cancers-12-03041-f001:**
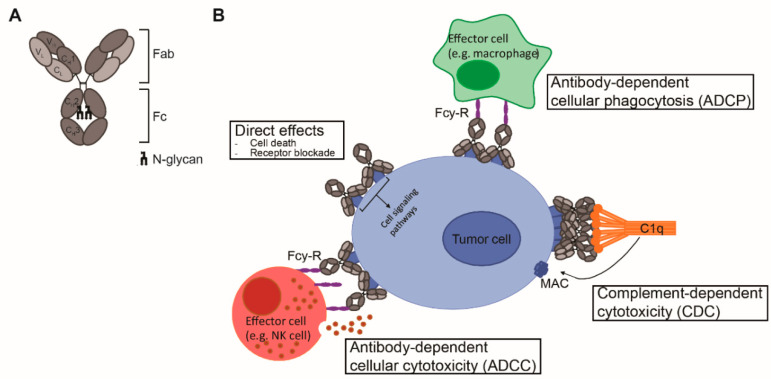
mAb structure and mechanisms of action. (**A**) Schematic representation of an IgG antibody consisting of variable (V) and constant (C) domains of the light (L; light grey) and heavy (H; dark grey) chains. The fragment antigen binding (Fab) domain is made up of V_L_ and C_L_ together with V_H_ and C_H_1, and the fragment crystallizable (Fc) region consists of C_H_2 and C_H_3, with an N-glycan attached to C_H_2. (**B**) Mechanisms of action of IgG1 mAbs consist of direct effects, including cell death and receptor blockade, and indirect effects, including antibody-dependent cellular cytotoxicity (ADCC), antibody-dependent cellular phagocytosis (ADCP) and complement-dependent cytotoxicity (CDC). ADCC and ADCP are mediated via binding to FcγR and cell death occurs via release of cytotoxic granules and via internalization and degradation of the target, respectively. CDC is mediated via binding to complement protein C1q and cell death occurs via formation of the membrane attack complex (MAC), which consists of complement proteins C5b, C6, C7, and C8 and various copies of C9, and generates pores in the membrane.

**Figure 2 cancers-12-03041-f002:**
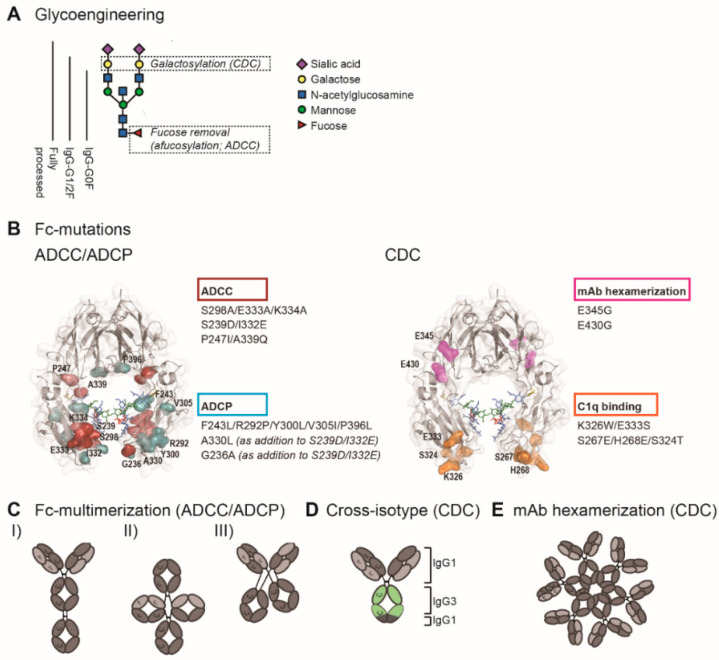
Fc engineering strategies to enhance Fc-effector functions. (**A**) The biantennary complex glycan structure of the most frequent variants for the N-glycan of therapeutic IgG (G0F, G1F, G2F, and fully processed). Glycoengineering strategies focus on galactosylation for enhanced CDC and afucosylation for enhanced ADCC. (**B**) ADCC/ADCP (left) and CDC (right) enhancing point mutations depicted in the crystal structure of the Fc region (PDB:3DO3). The ADCC-enhancing mutations S298A/E333A/K334A, S239D/I3332E, and P247I/A339Q are depicted in magenta; the ADCP-enhancing mutations F234L/R292P/Y300L/V305I/P396L, A330L, and G236A are shown in blue. A330L and G236A were added to the ADCC-enhancing mutations S239D/I332E to decrease the mutation-related increase in affinity to the inhibitory receptor FcγRIIb. The CDC-enhancing mutations E345G and E430G, which induce Fc/Fc interactions and subsequent mAb hexamerization (see **E**), are shown in orange and K326W/E333S and S267E/H268E/S324T, which enhance C1q binding affinity, are shown in pink. (**C**) Schematic representation of three Fc multimerization strategies which are aimed to enhance ADCC and ADCP. (**D**) Cross-isotype antibody generated by replacing the C_H_2 and part of the C_H_3 domains of an IgG1 antibody with the corresponding regions of an IgG3 antibody, to increase the CDC response. (**E**) Schematic representation of mAb hexamerization which facilitates C1q binding and enhances CDC, and can occur upon antigen binding by IgG1 antibodies naturally or when harboring the E345G or E430g mutations.

**Table 1 cancers-12-03041-t001:** Fc-engineered monoclonal antibodies (mAbs) with enhanced mAb effector function in (pre-) clinical development for chronic lymphocytic leukemia (CLL), B-cell non-Hodgkin lymphoma (B-NHL), and multiple myeloma (MM).

Fc-Engineered mAb	Target	Iso-Type	Chimeric/Human(ized)	Fc Engineering Strategy	Enhanced Effector Function	Additional mAb Engineering	Clinical Stage; NCT of Recruiting Clinical Trials	Major Indication(s)
Obinutuzumab (GA101;Gazyva)	CD20, type II	IgG2	Humanized	Afucosylation	ADCC	Modified elbow hinge	FDA-approved	FL and CLL
Ublituximab (LFB-R603, EMAB-6)	CD20, type I	IgG1	Chimeric	Low fucose	ADCC		Phase 2/3 ^1^	CLL and B-NHL
Ocaratuzumab (AME-133v, LY2469298)	CD20, type I	IgG1	Humanized	Mutations P247I/A339Q	ADCC	Antigen binding affinity optimized	Discontinued	
PRO131921 (RhuMAb; v114)	CD20, type I	IgG1	Humanized	Mutations (na)	ADCC and CDC		Discontinued	
Ocrelizumab	CD20	IgG1	Humanized	Mutations (na)	ADCC		Discontinued in hematology, approved for MS	MS
CD20 double engineered	CD20	IgG1		Afucosylation+ mutations S267E/H268F/S324T/G236A/I332E	ADCC and CDC		Preclinical	
	CD20	IgG1/IgG3		Afucosylation + mixed IgG1/IgG3 isotype	ADCC and CDC		Preclinical	
BI 836826	CD37	IgG1	Chimeric	Mutations S239D/I332E	ADCC		Discontinued	
DuoHexaBody-CD37 (GEN3009)	CD37	IgG1	Human	Mutation E430G (HexaBody)	CDC	Dual-epitope targeting	Phase 1 ^2^	B-NHL
Ianalumab (VAY736; B-1239)	BAFF-R	IgG1	Human	Afucosylation	ADCC		Phase 1 ^3^	CLL
Inebilizumab (MEDI-551)	CD19	IgG1κ	Humanized	Afucosylation	ADCC		Phase 1/2	B-cell malignancies
MDX-1342	CD19		Human	Afucosylation	ADCC and ADCP		Phase 1 halted	
Tafasitamab (MOR208, XmAb5574)	CD19	IgG1	Humanized	Mutations S239D/I332E	ADCC and ADCP	Antigen binding affinity optimized	Priority review granted by FDA	CLL and DLBCL
HexaBody-CD38 (GEN3014)	CD38	IgG1	Human	Mutation E430G (HexaBody)	CDC		Preclinical	B-NHL and MM
anti-CD38 SIFbody	CD38			Fc multimerization	ADCC and CDC		Preclinical	MM
XmAb5592	HM1.24	IgG1	Humanized	Mutations S239D/I332E				MM
	ICAM-1	IgG1	Human	Mutations S239D/I332E	ADCC and ADCP		Preclinical	MM
SEA-BCMA	BCMA	IgG1	Humanized	Afucosylation	ADCC and ADCP		Phase 1 ^4^	MM

^1^ ClinicalTrials.gov Identifier: NCT03828448; NCT04149821; NCT03801525; NCT02535286; NCT03379051; NCT04016805; NCT02793583. ^2^ ClinicalTrials.gov Identifier: NCT04358458. ^3^ ClinicalTrials.gov Identifier: NCT03400176. ^4^ ClinicalTrials.gov Identifier: NCT03582033.

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
