# Peer review of "Fc-Engineered Antibodies with Enhanced Fc-Effector Function for the Treatment of B-Cell Malignancies"

_cancers, 2020, doi:10.3390/cancers12103041_

Round 1
Reviewer 1 Report
This is a very thorough and excellent review highlighting the manner in which our advanced knowledge of Fc structure and function has bolstered the field of antibody-related therapies through advanced engineering methodologies. The review is highly informative, but not necessarily comprehensive since it focuses mostly on liquid tumor developments and not on solid tumors. But this does indeed render an exhaustive topic, manageable. The explanations of antibody structure, the various antibody functions, and the means utilizing Fc structure to obtain enhanced antibody activity are all well-presented. Also, the sections on enhanced antibody function and clinical development for various leukemias were very informative. Discussions of approaches, in terms of cancer target markers also was an excellent addition to the review.
There were only a few negative points.
A very important element of the various engineering approaches were overlooked and that is the impact of engineering on antibody stability/half-life/serum clearance. It would be helpful to add a section addressing this important issue, especially in light of the number of various approaches that are currently in clinic trial.
The review will be read by individuals with both academic and commercial interests. Thus, it would be helpful to have some idea of the manner in which these bioengineered products are produced (expressed and manufactured). Type of expression systems that are most prevalent. Problems in expression these proteins since some are stucturally complicated. It would be helpful to add a section that addresses this issue.
Regarding the clinical trial table, when relevant, it would be helpful to add Clinicaltrial.gov identifiers if applicable.
Reviewer 2 Report
This is a well written review of a potentially interesting topic, and therefore the target audience is relatively broad. On the positive side, there may be interest in the likely significant biological relevance for the treatment of B cells lymphoma. As described in the specific comments below, I think there are a few aspects of this topic that should be discussed in greater depth to give the reader a better idea of the complexities of this topic. On the negative side, while the topic is interesting, the molecular mechanisms of the Fc-engineering are ill-defined for receptor specificities and polymorphism. Despite
these limitations, the review may be useful for larger audiences.
Specific Comments:
- Increasing the affinity of Fc for FcRs is therefore of considerable interest. Unfortunately, the same cells also express the inhibitory FcRs receptor, whose extracellular domain shares significant homology with that of the activating FcRs like FcγRIIa shares about 93% homology with inhibitory FcγRIIb, and the task of selectively increasing the Fc affinity for the activating receptor remains a challenge which authors did not address in the manuscript.
- I would encourage authors to explain that how the Fc-engineering would help to address the issue of receptor polymorphism with multiple Ig isotypes.
- Several studies demonstrated that the presence of certain carbohydrate residues can lead to severely reduced ADCC efficiency. How Fc-engineering would help to control the nature of carbohydrates linked to the CH2 domain which has a major influence on the affinity of the Fc for FcRs.
Reviewer 3 Report
The cytotoxicity of humanized monoclonal antibody (mAb) such as antibody-dependent cellular cytotoxicity (ADCC), antibody-dependent cellular phagocytosis (ADCP), and complement-dependent cytotoxicity (CDC) mediated via the Fc domain of mAbs. In this paper, the authors summarized the novel approaches of engineering these Fc domains of mAbs to enhance their Fc-effector function and thereby their anti-tumor effects, with specific focus to comment their clinical status for the treatment against B lymphocytic malignancies. This review is useful for the readers of Cancers since it is concise and has lucid explanation and figures.
Author Response
We thank the reviewer for highlighting the relevance of this review for readers of Cancers.
Round 2
Reviewer 2 Report
Authors have addressed my commments, and improved the manuscript.